# Remnants of Galactic Subhalos and Their Impact on Indirect Dark-Matter Searches

**Martin Stref** [1,*] , **Thomas Lacroix** [2] **and Julien Lavalle** [1]

1 Laboratoire Univers et Particules de Montpellier (LUPM), Université de Montpellier & CNRS, Place Eugène Bataillon, 34095 Montpellier CEDEX 05, France; lavalle@in2p3.fr

2 Instituto de Fisica Teórica, C/ Nicolás Cabrera 13-15, Campus de Cantoblanco UAM, 28049 Madrid, Spain; thomas.lacroix@uam.es

* Correspondence: martin.stref@umontpellier.fr

**Abstract:** Dark-matter subhalos, predicted in large numbers in the cold-dark-matter scenario, should have an impact on dark-matter-particle searches. Recent results show that tidal disruption of these objects in computer simulations is overefficient due to numerical artifacts and resolution effects. Accounting for these results, we re-estimated the subhalo abundance in the Milky Way using semianalytical techniques. In particular, we showed that the boost factor for gamma rays and cosmic-ray antiprotons is increased by roughly a factor of two.

**Keywords:** particle dark matter; subhalos; indirect searches

## 1. Introduction

There is overwhelming evidence that most of the matter in the universe is nonbaryonic [1]. An exciting possibility to account for these puzzling observations is that the universe is filled with exotic particles that interact only very weakly with ordinary matter [2,3]. One of the most elegant and popular dark-matter (DM) particle candidates is the Weakly Interacting Massive Particle (WIMP). These hypothetical particles are being looked for in particle colliders [4–6], in direct detection experiments [7–9], and in cosmic radiation [10,11], so far without success. Although one of the motivations for WIMPs is related to the fact that they emerge naturally in particle theories addressing a hierarchy problem [12–14], WIMPs are also attractive stemming from their very simple thermal-production mechanism in the early universe. Moreover, a large fraction of available parameter space is still unconstrained and currently actively explored [15]. For completeness, it is worth recalling that many alternatives to WIMPs exist that we do not discuss here, like axions [16], sterile neutrinos [17], primordial black holes [18], and extended dark sectors [19]. The cosmological paradigm best supported by current probes is that DM is cold, i.e., collisionless and nonrelativistic. This implies a structuring of matter on scales smaller than typical galaxies, with a model-dependent cutoff [20,21]. Interestingly, subgalactic scales are those where there could be departures from the predictions of the cold DM paradigm because of some observational issues [22]. This might sign new specific properties of the dark matter (e.g., [23]), or it could be due to baryonic effects (e.g., [24]). This motivates a detailed inspection of the impact of DM properties on the smallest scales, irrespective of the underlying scenario.

The small-scale structuring of DM, as treated, for instance, in the WIMP scenario, translates into a large population of subhalos within galactic halos [25–27]. Modeling these subhalos is crucial if one is to make accurate predictions for direct and indirect DM searches. This is a difficult task, as numerical simulations are far from resolving the smallest structures predicted by the cold DM paradigm. To incorporate the smallest structures, one can extrapolate the results of simulations over orders of magnitude in scales (see, e.g., [28]) but this represents a leap of faith. On the other hand,

one can employ semianalytical models (see, e.g., [29–32]). The difficulty with the latter is accounting for the tidal effects experienced by subhalos within the host galaxy. These models can be calibrated on cosmological simulations, which are supposed to consistently describe the tidal stripping of subhalos in their host halo. However, it was recently pointed out by van den Bosch and collaborators [33,34] that simulations are plagued with numerical artifacts that lead to a significant overestimate of the tidal stripping efficiency, and therefore to an underestimate of the actual subhalo population even within the numerical resolution limit. An alternative and complementary way to study the tidal stripping of subhalos is to rely on analytical or semianalytical methods, which are based on first principles and allow to deal with subhalo mass scales, down to the free-streaming scale. Here, we review the semianalytical model developed by Stref and Lavalle [35] (SL17 hereafter), which incorporates a realistic and kinematically constrained Milky Way mass model (including baryons) and predicts the galactic subhalo abundance.[1] This model accounts for different sources of tidal effects, and can easily accommodate to different prescriptions for tidal disruption efficiency.

This paper is structured as follows. In Section 2, we briefly review the SL17 model and discuss the resilience of subhalos to tidal effects in light of recent analyses of simulation results [33,34]. In Section 3, we compute the DM mass density within subhalos, as well as the number density of these objects in the Milky Way. Finally, in Section 4, we look at the impact of our results on indirect searches for annihilating DM, focusing on gamma rays and cosmic-ray antiprotons.

## 2. Semianalytical Model of Galactic Subhalos

In this section, we review the SL17 Galactic subhalo population model and discuss the tidal effects experienced by subhalos. We then propose a way of incorporating the recent results of van den Bosch and collaborators in the model in a consistent calibration procedure.

### 2.1. Review of the Stref and Lavalle Model

SL17 is a semianalytical model of galactic subhalos that is built upon dynamical constraints and cosmological considerations. The main input of the model is the initial subhalo phase-space density

$$\frac{\mathrm{d}N}{\mathrm{d}V\,\mathrm{d}m\,\mathrm{d}c}(\vec{r}, m, c) \propto \frac{\mathrm{d}\mathcal{P}_v}{\mathrm{d}V}(\vec{r}) \times \frac{\mathrm{d}\mathcal{P}_m}{\mathrm{d}m}(m) \times \frac{\mathrm{d}\mathcal{P}_c}{\mathrm{d}c}(c, m),\tag{1}$$

where phase space refers to the position–mass–concentration space. Functions $\mathrm{d}\mathcal{P}_v/\mathrm{d}V$, $\mathrm{d}\mathcal{P}_m/\mathrm{d}m$ and $\mathrm{d}\mathcal{P}_c/\mathrm{d}c$ are the spatial, mass, and concentration distributions, respectively. It is assumed that, should subhalos behave as hard spheres (as is the case for single DM "particles" in a cosmological simulation), they would be spatially distributed as $\mathrm{d}\mathcal{P}_v/\mathrm{d}V \propto \rho_{\mathrm{DM}}$ where $\rho_{\mathrm{DM}}$ is the total DM density profile of the galaxy. This sets our initial conditions before tidal disruption. The smooth DM mass density is computed through

$$\rho_{\mathrm{sm}}(\vec{r}) = \rho_{\mathrm{DM}}(\vec{r}) - \langle \rho_{\mathrm{cl}} \rangle (\vec{r}),\tag{2}$$

where $\langle \rho_{\mathrm{cl}} \rangle$ is the average DM mass density inside clumps (this quantity is explicitly computed in Section 3). In the following, we use the galactic mass models constrained by McMillan [36] on pre-Gaia data for the DM and baryonic mass distributions. In this framework, Equation (2) ensures the compatibility of our subhalo model with the constrained DM profile $\rho_{\mathrm{DM}}$. Although this work is devoted to the study of Milky Way subhalos, SL17 can in principle be used to study the substructure population in any virialized DM system. One only needs a mass model for the system in question and a proper calibration of the subhalo mass fraction through the procedure outlined in Section 2.3.

---

[1]  We refer to this model as semianalytical because it involves integrals that must be computed numerically. The model does not rely on numerical simulations except at the level of a calibration described in Section 2.3.

Mass $m$ and concentration $c$ refer to cosmological mass $m_{200}$ and concentration $c_{200}$ (defined with respect to the critical density), where we dropped the 200 index for convenience. The subhalo mass function measured in simulations is consistent with a power law [26,27]

$$\frac{\mathrm{d}\mathcal{P}_m}{\mathrm{d}m}(m) \propto m^{-\alpha_m}\,\Theta(m - m_{\min})\,\Theta(m_{\max} - m)\,, \tag{3}$$

where $\Theta$ is the Heaviside step function, and the power-law index is $\alpha_m = 1.9$ or $\alpha_m = 2$. These values of $\alpha_m$ encompass the Press and Schechter [37] mass function and the Sheth and Tormen [38] mass function, as illustrated in Figure 1. These functions can be computed directly from the matter power spectrum in the framework of excursion set theory [39], for spherical collapse (Press–Schechter) and ellipsoidal collapse (Sheth–Tormen). Thus, the two power-law indices we considered bracket the theoretical uncertainties on the small-scale mass function. If the DM is made of WIMPs, mass cutoff $m_{\min}$ can be related to the kinetic decoupling of the DM particle and is found to lie between $10^{-4}\,\mathrm{M}_\odot$ and $10^{-10}\,\mathrm{M}_\odot$ [29,40–45]. Maximal mass $m_{\max}$ is set to $0.01 \times M_{\mathrm{DM}}$, where $M_{\mathrm{DM}}$ is the total DM mass in the Milky Way. Concentration distribution $\mathrm{d}\mathcal{P}_c/\mathrm{d}c$ is generically found to exhibit a log-normal distribution for field halos [46,47], which defines our initial concentration distribution (before tidal stripping). We adopt the peak value and variance fit in Sánchez-Conde and Prada [48], which was shown to provide a good description of cosmological simulations run independently by several groups. Subhalos are assumed to have a Navarro–Frenk–White (NFW) profile [49] with parameters set by $m$ and $c$ (the impact of choosing an Einasto profile [50] instead of NFW was investigated in [35]).

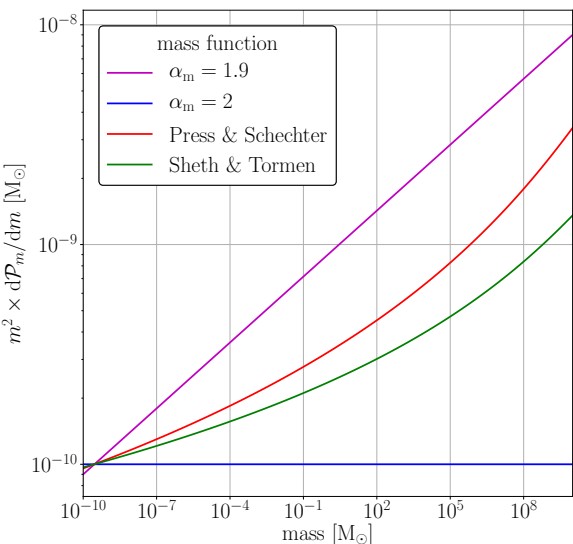

**Figure 1.** Mass function $\mathrm{d}n/\mathrm{d}m$ multiplied by $m^2$. We show the prediction of Press and Schechter [37] (red line), Sheth and Tormen [38] (green line) as well as the power-law mass functions with index $\alpha_m = 1.9$ (magenta line) and $\alpha_m = 2$ (blue line). The Press–Schechter and Sheth–Tormen mass functions were computed for the cosmology of Planck 2018 [1] using the transfer function of Eisenstein and Hu [51] and a sharp-$k$ filter. All mass functions are normalized to unity with $M_{\min} = 10^{-10}\,\mathrm{M}_\odot$.

The subhalo population is strongly affected by tidal interactions with the potential of the host galaxy [52]. This is accounted for in the model through the calculation of a tidal radius $r_{\mathrm{t}}$ for each subhalo. The tidal radius should be interpreted as the physical extension of a subhalo, which is in general smaller than the extension it would have on a flat background. The physical mass of a subhalo is then

$$m_{\mathrm{t}}(\vec{r}, m, c) = 4\pi \int_0^{r_{\mathrm{t}}(\vec{r}, m, c)} \mathrm{d}x\, x^2\, \rho_{\mathrm{sub}}(x) \leq m\,, \tag{4}$$

where $\rho_{\text{sub}}$ is the subhalo inner mass density profile. In our modeling, tidal stripping only removes the outer layers of subhalos while leaving the inner parts unchanged. In reality, DM should rearrange itself into a new equilibrium state. Central density however, should be left essentially unchanged [53]. Since we are interested in indirect searches for annihilating DM, and the central density gives the dominant contribution to the annihilation rate, our modeling should lead to a reasonable approximation. Two important contributions are accounted for in SL17: the effect of the smooth galactic potential (including both DM and baryons), and the gravitational shocking induced by the baryonic disk [54,55]. The latter effect turns out to be very efficient at stripping subhalos in the inner 20 kpc of the galaxy, a result also found in numerical studies [56–58]. The strength of SL17 over simulations is that it accounts for the constrained potential of the MW, with a detailed description of baryons.

*2.2. Subhalo Disruption?*

Whether a subhalo can be completely disrupted by tidal effects is an open question. A number of numerical studies found that a subhalo is completely disrupted when the total energy gained through tidal-stripping or disk-shocking effects is comparable to the binding energy [56,59]. On the other hand, some studies [60–62] found that cuspy subhalos almost always survive mass loss, leaving a small bound remnant behind even after gaining an energy far greater than their binding energy. These contradictory results may have been reconciled in a recent series of papers by van den Bosch and collaborators [33,34,63]. In these studies, it is shown that subhalo disruption in N-body simulations can actually be entirely explained by numerical artifacts. In particular, disruption is shown to be highly sensitive to the value of the force-softening length. If this length is taken sufficiently small, the authors showed that subhalos survive tidal mass loss in the form of a small bound remnant. We aim at quantifying the impact of these results on the whole subhalo population. Tidal disruption is modeled in a very simple way in SL17: given a subhalo with scale radius $r_{\text{s}}$ and tidal radius $r_{\text{t}}$, we assume

$$\frac{r_{\text{t}}(\vec{r}, m, c)}{r_{\text{s}}(m, c)} < \epsilon_{\text{t}} \Leftrightarrow \text{subhalo is disrupted} \tag{5}$$

In Equation (5), $\epsilon_{\text{t}}$ is a dimensionless free parameter assumed universal, i.e., independent of the subhalo's mass, concentration, or position. In SL17, the value of the disruption parameter was set to $\epsilon_{\text{t}} = 1$ in agreement with numerical results (see, e.g., [59]). The results of van den Bosch and collaborators point toward a much lower value for $\epsilon_{\text{t}}$. In this work, we consider two extreme values: $\epsilon_{\text{t}} = 1$ and $\epsilon_{\text{t}} = 0.01$. The latter means a subhalo is disrupted when it has lost around 99.99% of its mass. In the following, we refer to these two configurations as "fragile subhalos" ($\epsilon_{\text{t}} = 1$) and "resilient subhalos" ($\epsilon_{\text{t}} = 0.01$). The final subhalo phase-space density can now be written:

$$\frac{\text{d}N}{\text{d}V\,\text{d}m\,\text{d}c}(\vec{r}, m, c) = \frac{N_{\text{tot}}}{K_{\text{tot}}} \frac{\text{d}\mathcal{P}_v}{\text{d}V}(\vec{r}) \times \frac{\text{d}\mathcal{P}_m}{\text{d}m}(m) \times \frac{\text{d}\mathcal{P}_c}{\text{d}c}(c, m)\, \Theta\left(\frac{r_{\text{t}}(\vec{r}, m, c)}{r_{\text{s}}(m, c)} - \epsilon_{\text{t}}\right), \tag{6}$$

where $N_{\text{tot}}$ is the total number of substructures within the virial radius of the Milky Way, and $K_{\text{tot}}$ is a normalization factor:

$$K_{\text{tot}} = \int \text{d}V\, \frac{\text{d}\mathcal{P}_v}{\text{d}V}(\vec{r}) \int \text{d}m\, \frac{\text{d}\mathcal{P}_m}{\text{d}m}(m) \int \text{d}c\, \frac{\text{d}\mathcal{P}_c}{\text{d}c}(c, m)\, \Theta\left(\frac{r_{\text{t}}(\vec{r}, m, c)}{r_{\text{s}}(m, c)} - \epsilon_{\text{t}}\right). \tag{7}$$

*2.3. Calibration Procedure*

In its current version, the SL17 model requires a calibration of the subhalo abundance in a given mass range (this will change in future versions). To be consistent with results from the highest-resolution simulations available, calibration is done by demanding that the subhalo mass fraction is similar to what is found in the dark matter-only Via Lactea II simulation [26]. This amounts to 11% of the total dark halo mass in the form of subhalos in the virial mass range $[m_1, m_2] = [2.2 \times 10^{-6} M_{\text{DM}}, 8.8 \times 10^{-4} M_{\text{DM}}]$ where $M_{\text{DM}}$ is the total DM mass of the galaxy. We stress

that these numbers are expressed in terms of virial masses, not tidal masses (see [35] for further details). To reproduce the (likely overestimated) tidal disruption efficiency in simulations, we have to set $\epsilon_t = 1$ at the calibration stage, and we also neglect the impact of baryons. Disruption efficiency parameter $\epsilon_t$ can safely be changed after the calibration has been completed. It is much safer to perform this calibration on dark-matter-only simulations because tidal stripping induced by baryons strongly depends on the details of the stellar distribution, which is acutely constrained in the Milky Way. More formally, the normalization procedure reads:

$$f_{\text{sub}}(m_1, m_2) = \frac{1}{M_{\text{DM}}} \int dV \int_{m_1}^{m_2} dm \int dc \times m \times \left. \frac{dN}{dV \, dm \, dc} \right|_{\text{DMO}, \epsilon_t = 1}. \tag{8}$$

Fixing $f_{\text{sub}}(m_1, m_2) = 0.11$ leads to the total number of clumps $N_{\text{DMO}, \epsilon_t = 1}$ in the simulation-like configuration. Note that this value assumes that $m$ is really $m_{200}$ in the equation above, not the tidal mass.

Now that the model is properly calibrated, we incorporate all the effects that are not included in the calibration, i.e., the tidal effects due to the baryons and possibly $\epsilon_t < 1$. This is done by assuming that subhalos in the outskirts of the galaxy are not affected by baryonic tides or the value of $\epsilon_t$. This is motivated by the observation that tidal effects are inefficient far from the center of the galaxy, and subhalos almost behave like isolated halos. The DM mass within clumps per unit of volume can be expressed as

$$\langle \rho_{\text{cl}} \rangle (\vec{r}) = \int_{m_{\text{min}}}^{m_{\text{max}}} dm \int_1^{\infty} dc \, \frac{dN}{dV \, dm \, dc} \, m_t(\vec{r}, m, c), \tag{9}$$

where $m_t$ is the tidal subhalo mass introduced in Equation (4). Equating $\langle \rho_{\text{cl}} \rangle (r_{200})$, where $r_{200}$ is the virial radius of the galaxy, in the DM-only + $\epsilon_t = 1$ configuration, to the same quantity in the realistic configuration (including baryons and $\epsilon_t \leq 1$) leads to the simple relation

$$\frac{N_{\text{DMO}, \epsilon_t = 1}}{K_{\text{DMO}, \epsilon_t = 1}} = \frac{N_{\text{tot}}}{K_{\text{tot}}}. \tag{10}$$

The two normalization factors $K$ can be computed using Equation (7), and we obtain the value of $N_{\text{tot}}$. The number of subhalos within the solar radius $r_{\odot} = 8.21$ kpc is shown in Table 1. This number is highly sensitive to the parameters of mass function $\alpha_m$ and $m_{\text{min}}$, as already shown in Reference [35]. Furthermore, it is quite sensitive to $\epsilon_t$: going from $\epsilon_t = 1$ to $\epsilon_t = 0.01$, the number of subhalos increases by at least an order of magnitude. The impact of $\epsilon_t$ on subhalo mass and number density is investigated in the next section.

**Table 1. Top panel:** number of subhalos within $r_{\odot} = 8.21$ kpc, for different values of mass function parameters $\alpha_m$ and $m_{\text{min}}$, for $\epsilon_t = 1$. **Bottom panel:** same as top panel, for $\epsilon_t = 0.01$.

| $\epsilon_t = 1$ | $m_{\text{min}} = 10^{-4} \, M_{\odot}$ | $m_{\text{min}} = 10^{-10} \, M_{\odot}$ |
|---|---|---|
| $\alpha_m = 1.9$ | $1.90 \times 10^{10}$ | $1.55 \times 10^{16}$ |
| $\alpha_m = 2$ | $2.64 \times 10^{11}$ | $8.40 \times 10^{17}$ |
| $\epsilon_t = 0.01$ | $m_{\text{min}} = 10^{-4} \, M_{\odot}$ | $m_{\text{min}} = 10^{-10} \, M_{\odot}$ |
| $\alpha_m = 1.9$ | $6.64 \times 10^{11}$ | $1.68 \times 10^{17}$ |
| $\alpha_m = 2$ | $9.06 \times 10^{12}$ | $9.10 \times 10^{18}$ |

## 3. Mass and Number Densities of Subhalos

In this section, we computed the mass density within subhalos, as well as the subhalo number density. Subhalo mass density is defined in Equation (9). Once subhalo density is known, it is used to

determine the amount of DM smoothly distributed across the galaxy through Equation (2). The DM mass inside subhalos was compared with the total DM density $\rho_{\text{DM}}$ in Figure 2. Mass density in the form of subhalos is predicted to be much higher, by orders of magnitude, for resilient subhalos than for fragile subhalos. The former case is also more theoretically justified, although the latter one allows us to compare with very conservative assumptions. At the position of the Solar System, the impact is around one order of magnitude. Although these are large differences, we note the subhalo mass density is still far below the total DM density. This means that most of the DM mass within the orbit of the Sun is smoothly distributed rather than clumpy, irrespective of the efficiency of the tidal disruption set by $\epsilon_{\text{t}}$.

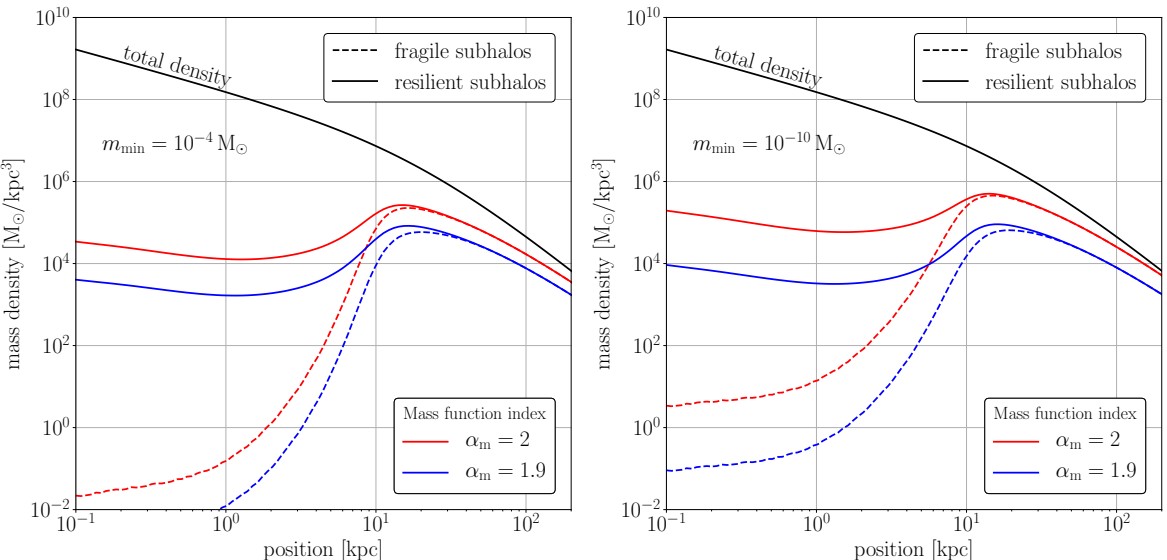

**Figure 2. Left panel:** Dark matter (DM) mass density inside subhalos $\langle \rho_{\text{cl}} \rangle$ for $m_{\text{min}} = 10^{-4}\,\text{M}_\odot$. The mass function index is $\alpha_{\text{m}} = 2$ (red) or $\alpha_{\text{m}} = 1.9$ (blue). We show the result for $\epsilon_{\text{t}} = 1$ (dashed) and $\epsilon_{\text{t}} = 0.01$ (solid). Total DM density is shown as a black solid curve for comparison. **Right panel:** same as left panel, for $m_{\text{min}} = 10^{-10}\,\text{M}_\odot$.

The subhalo number density in SL17 can be formally written:

$$\frac{\text{d}N}{\text{d}V}(\vec{r}) = \int \text{d}m \int \text{d}c\, \frac{\text{d}N}{\text{d}V\,\text{d}m\,\text{d}c} \tag{11}$$

$$= \frac{N_{\text{tot}}}{K_{\text{tot}}} \frac{\text{d}\mathcal{P}_v}{\text{d}V}(\vec{r}) \int_{m_{\text{min}}}^{m_{\text{max}}} \text{d}m\, \frac{\text{d}\mathcal{P}_m}{\text{d}m}(m) \int_1^\infty \text{d}c\, \frac{\text{d}\mathcal{P}_c}{\text{d}c}(c,m)\, \Theta\left( \frac{r_{\text{t}}(\vec{r},m,c)}{r_{\text{s}}} - \epsilon_{\text{t}} \right). \tag{12}$$

The obtained results are shown in Figure 3. Number density, just like mass density, is highly sensitive to $\alpha_{\text{m}}$ and $m_{\text{min}}$, as well as the disruption parameter. Interestingly, the values we get in the Solar neighborhood are comparable to the local number density of stars $n_* \sim 1\,\text{pc}^{-3}$. For a low value of minimal mass $m_{\text{min}}$, the subhalo number density can even be much higher, possibly going as high as $10^5\,\text{pc}^{-3}$. This could have a number of interesting implications for the interactions between subhalos and stars. The tidal heating of subhalos by stars has been investigated in a number of studies [29,60,64–68], with different conclusions. In the next section, we look at the impact of our results on indirect DM searches.

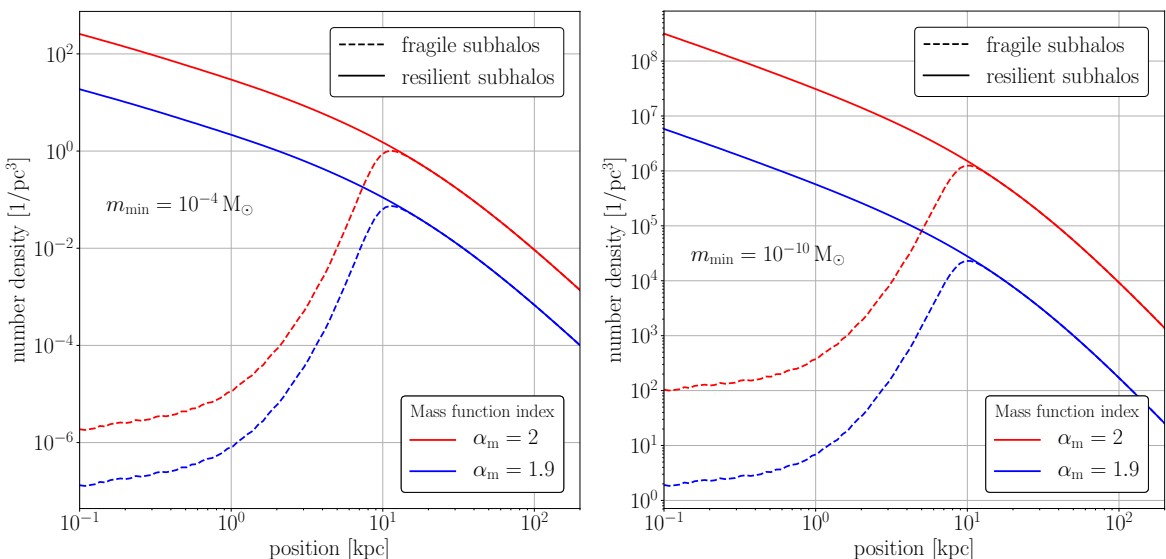

**Figure 3. Left panel:** subhalo number density for $m_{\min} = 10^{-4}\,\mathrm{M}_\odot$. Mass function index is $\alpha_{\mathrm{m}} = 2$ (red) or $\alpha_{\mathrm{m}} = 1.9$ (blue). We show the result for $\epsilon_{\mathrm{t}} = 1$ (dashed) and $\epsilon_{\mathrm{t}} = 0.01$ (solid). **Right panel:** same as left panel, for $m_{\min} = 10^{-10}\,\mathrm{M}_\odot$.

## 4. Impact on Indirect Searches for Annihilating Dark Matter

In this section, we quantify the impact of galactic clumps on indirect searches for self-annihilating DM. Inhomogeneities are known to enhance the DM annihilation rate in galactic halos [69]. We computed the local DM self-annihilation rate and evaluated the enhancement due to the survival of clump remnants, referred to as the boost factor. Two complementary channels were then investigated: gamma rays and antiproton cosmic rays.

### 4.1. Annihilation Profiles and Local Boost Factors

The number of self-annihilation of DM particles at position $\vec{r}$ is proportional to $\rho^2(\vec{r})$, where $\rho$ is DM mass density. If subhalos are discarded, the galactic annihilation profile is

$$\mathcal{L}_0(\vec{r}) = \rho_{\mathrm{DM}}^2(\vec{r})\,. \tag{13}$$

Let us now consistently include the contribution of subhalos. The luminosity of a single clump is

$$L_{\mathrm{t}}(\vec{r}, m, c) = \int_{V_{\mathrm{t}}} \mathrm{d}^3\vec{r}\, \rho_{\mathrm{sub}}^2(\vec{r})\,, \tag{14}$$

where $V_{\mathrm{t}}(\vec{r}, m, c)$ is the volume of the clump within its tidal radius. The annihilation of the full subhalo population is simply obtained by integrating the luminosity of a single object over the subhalo phase-space number density:

$$\mathcal{L}_{\mathrm{cl}}(\vec{r}) = \int_{m_{\min}}^{m_{\max}} \mathrm{d}m \int_1^{+\infty} \mathrm{d}c\, \frac{\mathrm{d}N}{\mathrm{d}V\mathrm{d}m\mathrm{d}c}\, L_{\mathrm{t}}(\vec{r}, m, c)\,. \tag{15}$$

The full annihilation profile must also incorporate the annihilation in the smooth halo (different from $\mathcal{L}_0$, which is the density assuming all the DM is smoothly distributed), as well as the annihilation of subhalo particles onto smooth halo particles. The first contribution can be written as:

$$\mathcal{L}_{\mathrm{sm}}(\vec{r}) = \rho_{\mathrm{sm}}^2(\vec{r})\,, \tag{16}$$

where $\rho_{\rm sm}(\vec{r}) = \rho_{\rm DM}(\vec{r}) - \langle \rho_{\rm cl} \rangle (\vec{r})$ is the smooth DM density. The clump-smooth contribution is:

$$\mathcal{L}_{\rm cs}(\vec{r}) \;=\; 2\,\rho_{\rm sm}(\vec{r})\,\langle \rho_{\rm cl} \rangle (\vec{r}) \tag{17}$$

$$=\; 2\,\rho_{\rm sm}(\vec{r}) \int_{m_{\rm min}}^{m_{\rm max}} {\rm d}m \int_{1}^{+\infty} {\rm d}c\, \frac{{\rm d}N}{{\rm d}V {\rm d}m {\rm d}c}\, m_{\rm t}(\vec{r}, m, c)\,. \tag{18}$$

The total annihilation profile is simply the sum of all contributions:

$$\mathcal{L}(\vec{r}) = \mathcal{L}_{\rm cl}(\vec{r}) + \mathcal{L}_{\rm sm}(\vec{r}) + \mathcal{L}_{\rm cs}(\vec{r})\,. \tag{19}$$

This should be compared to $\mathcal{L}_0(\vec{r})$ to evaluate the impact of clustering on the annihilation rate. This is usually done in terms of a boost factor, which we define as

$$1 + \mathcal{B}(\vec{r}) = \frac{\mathcal{L}(\vec{r})}{\mathcal{L}_0(\vec{r})}\,. \tag{20}$$

This is not quite the boost factor used in indirect searches, which is defined through a ratio of fluxes (see Equations (23) and (29)). The boost in Equation (20) is rather the local increase in the annihilation rate due to clustering. According to this definition, the boost is zero if $\mathcal{L} = \mathcal{L}_0$, i.e., substructures are not included.[2]

The annihilation profiles are shown on the top panels in Figure 4, and the associated boost factors are shown on the bottom panels. As already shown a long time ago, see e.g., References [29,70], the boost is an increasing function of the galactocentric radius $r = |\vec{r}|$. This is due to the morphology of the annihilation profiles that is modified by the inclusion of clumps: we have $\mathcal{L}_{\rm cl} \propto \rho_{\rm DM}$ while $\mathcal{L}_0 \propto \rho_{\rm DM}^2$. The high sensitivity of the annihilation profile to the mass function index $\alpha_{\rm m}$ is also noticeable, which is by far the largest source of uncertainty on clump contribution. The value of disruption parameter $\epsilon_{\rm t}$ has almost no impact on the boost above 20 kpc due to the ineffectiveness of tidal effects far from the center of the galaxy. Below 20 kpc, the boost is strongly sensitive to the disruption parameter. In the inner few kiloparsecs, fixing $\epsilon_{\rm t} = 0.01$ leads to a boost orders of magnitude larger than in the $\epsilon_{\rm t} = 1$ configuration. We, however, have $\mathcal{B}(\vec{r}) \ll 1$ below 3 kpc regardless of the value of $\epsilon_{\rm t}$, meaning $\mathcal{L} \simeq \mathcal{L}_0$ and subhalos do not have any impact on the annihilation rate in that region. The region where the impact of $\epsilon_{\rm t}$ on the annihilation profile is the most important is located between 3 and 10 kpc. This region coincidentally includes the Solar System, located at $r \simeq 8$ kpc. This motivates a more detailed investigation of two standard annihilation channels: gamma rays and cosmic-ray antiprotons, which are sensitive to different annihilation regions.

---

[2]  This differs by one unit from the definition used in Reference [35].

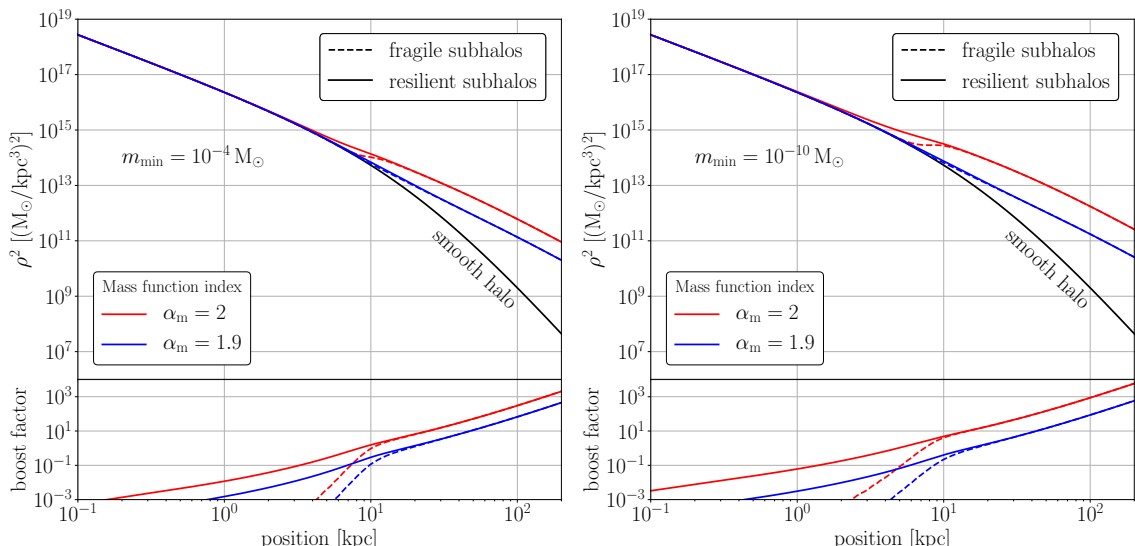

**Figure 4.** Total luminosity density profiles as defined in Equation (19), for a mass function index $\alpha_m = 2$ (red) and $\alpha_m = 1.9$ (blue). We show the results for efficient tidal disruption ($\epsilon_t = 1$, dashed) and very resilient clumps ($\epsilon_t = 0.01$, solid). (Left panel) Results for $m_{min} = 10^{-4} \, M_\odot$, (right panel) results for $m_{min} = 10^{-10} \, M_\odot$. The total luminosity density without clumps is displayed as a solid black line on each panel.

### 4.2. Application to Gamma Rays

The energy-differential flux of gamma rays originating from DM self-annihilation is, on a given line of sight,

$$\frac{d\Phi_\gamma}{dE \, d\Omega} = \frac{1}{4\pi} \frac{\langle \sigma v \rangle}{2 \, m_\chi^2} \frac{dN_\gamma}{dE} \int ds \, \rho^2 \,, \tag{21}$$

where $\langle \sigma v \rangle$ is the thermally averaged annihilation cross-section, $m_\chi$ is the DM mass, $dN_\gamma/dE$ is the gamma-ray spectrum at annihilation, and $s$ the distance coordinate along the line of sight.[3] If the annihilation cross-section is velocity-independent, astrophysical ingredients only enter through the J-factor, defined as

$$J(\psi) = \int ds \, \rho^2(s, \psi) \,, \tag{22}$$

where $\psi$ is the angle between the direction of the galactic center and the line of sight (spherical symmetry of the dark halo is assumed). The impact of small-scale clustering on this J-factor has been considered in a number of studies [32,48,70–78]. We define the gamma-ray boost factor as

$$1 + \mathcal{B}_\gamma(\psi) = \frac{J(\psi)}{J_0(\psi)} \,, \tag{23}$$

where $J_0 = \int ds \, \mathcal{L}_0$ is the J-factor without subhalos. Unlike local boost $\mathcal{B}$ in Equation (20), gamma-ray boost $\mathcal{B}_\gamma$ depends on line of sight rather than the position in the galaxy [70]. The boost is shown as a function of $\psi$ in Figure 5. The growth of local boost $\mathcal{B}(\vec{r})$ as a function of $r$ translates into a growth of $\mathcal{B}_\gamma(\psi)$ as a function of $\psi$, i.e., the maximal gamma-ray boost is reached at the anticenter. This maximal boost ranges from 0.5 to 9, depending on the values of $\alpha_m$ and $m_{min}$. The survival of clumps noticeably increases the boost at all latitudes. The gain is greater at small latitudes where substructures are more

---

3    The expression of $d\Phi_\gamma/dE$ should be multiplied by $1/2$ if the DM particle is not its own antiparticle.

impacted by tidal effects. Below $\psi \simeq 40$ deg, the boost is increased by a factor of at least two in all configurations. This should have important consequences for indirect searches using gamma rays, especially at high latitudes. Interestingly, high latitudes have been shown to be a very sensitive probe of DM annihilation even without the inclusion of clumps [79].

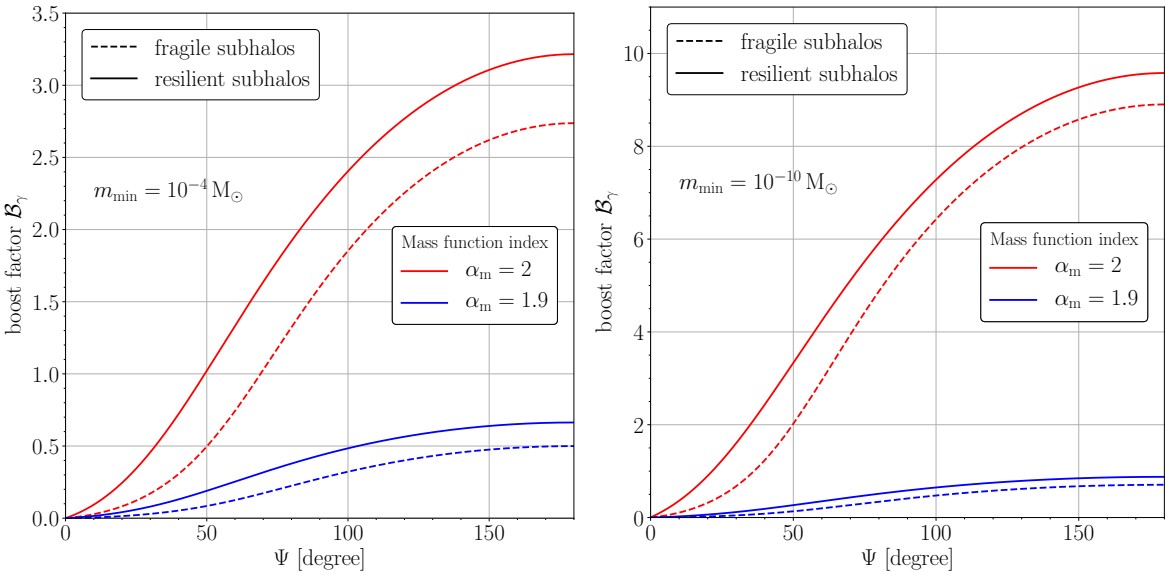

**Figure 5. Left panel:** gamma-ray boost factor as defined in Equation (23), as a function of angle $\psi$ between the direction of the galactic center and the line of sight, for a minimal subhalo mass of $m_{\min} = 10^{-4}\,\mathrm{M_\odot}$. We show the results for efficient tidal disruption ($\epsilon_t = 1$, dashed) and very resilient clumps ($\epsilon_t = 0.01$, solid). **Right panel:** same as left panel, for $m_{\min} = 10^{-10}\,\mathrm{M_\odot}$.

### 4.3. Application to Cosmic-Ray Antiprotons

Charged cosmic rays constitute an indirect detection channel, complementary to gamma rays [10,80]. Since their original proposal as a probe of DM annihilation [81], cosmic-ray antiprotons have been shown to be especially sensitive, see e.g., References [82–88]. Antiprotons have been the subject of much scrutiny since the latest measurement of the antiproton flux performed by the AMS-2 collaboration [89]. A number of studies [90–95] have found a discrepancy between the measured flux and a purely secondary origin of antiprotons. This discrepancy could be interpreted as evidence for annihilating DM, although the significance of the excess is debated as it depends on the propagation model used and the modeling of systematic uncertainties. In this context, it is worth evaluating systematic uncertainties coming from small-scale structuring.

Since antiprotons have a random motion due to their diffusion on the inhomogeneities of the magnetic halo, their detection gives little information on their source. This implies that the antiproton boost factor, and the boost for charged cosmic rays in general, is not direction-dependent, unlike for gamma rays. Instead, this boost is energy-dependent [96] and has been shown to be mild, at most a factor of two [31,74]. Although smaller than the gamma-ray boost, this can still be larger than the systematic uncertainties on cosmic-ray propagation. This motivates a new computation of the boost, which we performed here. The antiproton boost factor is defined as:

$$1 + \mathcal{B}_{\bar{p}}(T) = \frac{\Phi_{\bar{p}}(T)}{\Phi_{\bar{p},0}(T)}, \qquad (24)$$

where $T$ is the antiproton kinetic energy, $\Phi_{\bar{p}}$ is the DM-induced antiproton flux including the subhalo contribution, and $\Phi_{\bar{p},0}$ the same flux assuming all the DM is smoothly distributed. To obtain the flux, one must solve the cosmic-ray steady-state propagation equation

$$-K\Delta\Psi + \vec{\nabla}.(\vec{V}_c\Psi) + \partial_E\left[b\,\Psi - K_{EE}\partial_E\Psi\right] + 2h\delta(z)\,\Gamma_{\mathrm{ann}}\Psi = Q_{\mathrm{DM}}, \tag{25}$$

which accounts for spatial diffusion, convection, energy losses, diffusive re-acceleration, and spallation processes in the disk (taken as infinitely thin). In Equation (25), $\Psi$ is the antiproton number density per unit energy that is related to the flux through $\Phi_{\bar{p}} = v_{\bar{p}}/(4\pi) \times \Psi$ where $v_{\bar{p}}$ is the antiproton speed. Antiprotons are sourced by DM annihilation:

$$Q_{\mathrm{DM}}(E,\vec{r}) = \frac{\langle\sigma v\rangle}{2}\frac{\mathrm{d}N_{\bar{p}}}{\mathrm{d}E}\left(\frac{\rho(\vec{r})}{m_\chi}\right)^2, \tag{26}$$

where $\mathrm{d}N_{\bar{p}}/\mathrm{d}E$ is the antiproton spectrum at annihilation.[4] Several unknown propagation parameters enter Equation (25). These can be constrained using the measured boron-to-carbon ratio (B/C) [97]. We used the best-fit model derived by Reinert and Winkler [92], which includes an energy break in the diffusion coefficient. The B/C ratio can only constrain $K_0/L$, where $K_0$ is the normalization of the diffusion coefficient, and $L$ is the half-height of the magnetic halo. As shown in Reference [83], the DM-induced antiproton flux crucially depends on $L$; hence, we considered two extremal values in this work. A lower bound on $L$ can be obtained from low-energy positron data [98], and the authors of Reference [92] found $L = 4.1\,\mathrm{kpc}$. For the largest value, we took $L = 15\,\mathrm{kpc}$. According to the analysis of Reference [92], the B/C data are consistent with negligible re-acceleration. Furthermore, we neglected energy losses that are unimportant for high-energy antiprotons. The resulting transport equation can be solved semianalytically using Green's function formalism (see Reference [31] for the solution), and the differential flux can be written as:

$$\frac{\mathrm{d}\Phi_{\bar{p}}}{\mathrm{d}T\,\mathrm{d}\Omega} = \frac{v_{\bar{p}}}{4\pi}\frac{\langle\sigma v\rangle}{2\,m_\chi^2}\int \mathrm{d}E_s\int \mathrm{d}^3\vec{r}_s\,G(E \leftarrow E_s; \vec{r}_\odot \leftarrow \vec{r}_s)\frac{\mathrm{d}N_{\bar{p}}}{\mathrm{d}E}(E_s)\,\rho^2(\vec{r}_s). \tag{27}$$

Since all energy-dependent terms have been neglected in Equation (25), the energy part of Green's function is trivial

$$G(E \leftarrow E_s; \vec{r}_\odot \leftarrow \vec{r}_s) = \delta(E - E_s) \times \overline{G}(\vec{r}_\odot \leftarrow \vec{r}_s), \tag{28}$$

and the boost factor can be simply written:

$$1 + \mathcal{B}_{\bar{p}}(T) = \frac{\int \mathrm{d}^3\vec{r}_s\,\overline{G}(\vec{r}_\odot \leftarrow \vec{r}_s)\,\mathcal{L}(\vec{r}_s)}{\int \mathrm{d}^3\vec{r}_s\,\overline{G}(\vec{r}_\odot \leftarrow \vec{r}_s)\,\mathcal{L}_0(\vec{r}_s)}. \tag{29}$$

The boost factor is shown as a function of the antiproton kinetic energy in Figure 6. We first note that the boost is roughly energy-independent. This is because antiprotons probe the entire volume of the magnetic halo during their lifetime, independently of their energy. This is not true at low energies, below a few GeVs, where energy losses become relevant. The half-height of the magnetic halo has a small impact on the boost, with $L = 15\,\mathrm{kpc}$ leading to a slightly larger $\mathcal{B}_{\bar{p}}$ than $L = 4.1\,\mathrm{kpc}$. The main source of uncertainties are coming from subhalo parameters $\alpha_m$, $m_{\mathrm{min}}$, and $\epsilon_t$. Survival parameter $\epsilon_t$ had significant impact on the result, with a small value $\epsilon_t = 0.01$ leading to a boost roughly twice as large as in the $\epsilon_t = 1$ case. As for gamma rays, the most critical parameter is $\alpha_m$. For a low-value $\alpha_m = 1.9$, the boost never exceeded 10%, while it was always higher for $\alpha_m = 2$.

---

[4]    If DM is not its own antiparticle, $Q$ should be divided by 2.

Overall, playing with the propagation and subhalo parameters, we found that the antiproton boost can conservatively range from 2% to 140%. These values are in agreement with earlier results. Although it is conservative to ignore small-scale clustering when deriving limits on the annihilation cross-section using data, the boost should be included when interpreting an excess as a signature of DM annihilation. Indeed, a factor of two in the DM contribution would change the inferred mass and cross-section of the hypothetical DM particle.

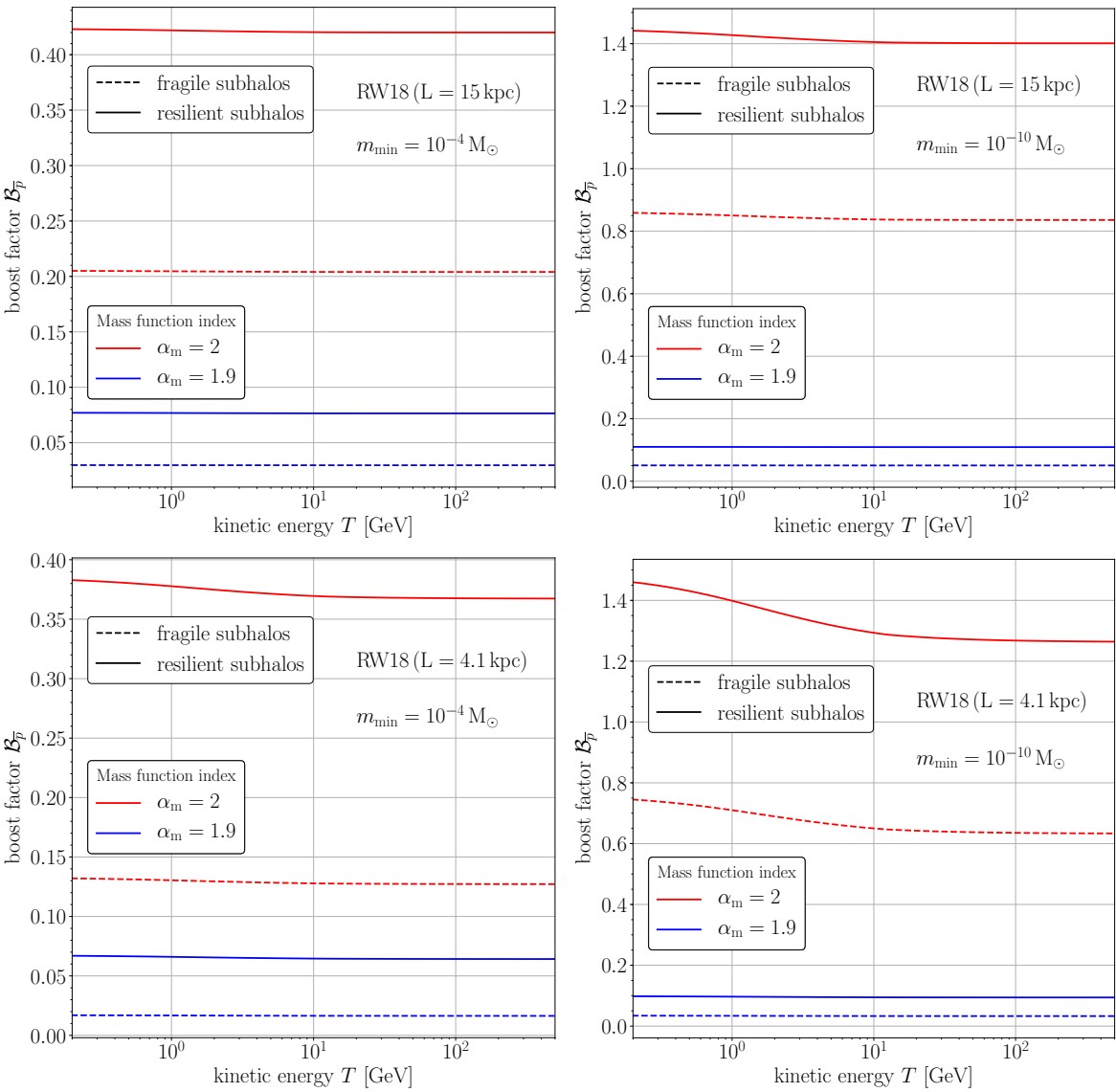

**Figure 6. Top panel:** antiproton boost factor as a function of kinetic energy, as defined in Equation (29). We show the result for a half-height of the magnetic halo of $L = 15\,\mathrm{kpc}$, and a minimal subhalo mass of $m_{\mathrm{min}} = 10^{-4}\,\mathrm{M_\odot}$ (left) or $m_{\mathrm{min}} = 10^{-10}\,\mathrm{M_\odot}$ (right). **Bottom panel:** same as top panel, for half-height $L = 4.1\,\mathrm{kpc}$.

## 5. Conclusions

Subhalos suffer mass loss due to their interaction with the tidal field of the galaxy, which makes their modeling very challenging. Consequently, most subhalo models rely at least partly on numerical simulations to calibrate their predictions. However, it was recently shown that numerical simulations might not properly account for the tidal disruption of subhalos, as artificial effects lead to a serious overestimation of the efficiency of these processes [33,34]. We note that the resistance of subhalos to tidal stripping is further supported by theoretical arguments, like adiabatic invariance that should

prevail in their inner parts [55], as already emphasized in Reference [35]. We derived some of the consequences of these results using the semianalytical galactic subhalo population model of Stref and Lavalle [35], assuming tidal disruption efficiency $\epsilon_t = 0.01$. We predicted the spatial dependence of the subhalo properties due to tides induced both by the global gravitational potential and baryonic disk shocking. We remind the reader that this model is built from constrained mass models for the Milky Way and is therefore consistent with current kinematic constraints, which is usually not the case in extrapolations from "Milky Way-like" simulations. We found that the local mass density is still dominated by the smooth component of the dark halo. The local number density of subhalos is increased by roughly one order of magnitude with respect to estimates based on a tidal disruption efficiency similar to that inferred from simulations ($\epsilon_t = 1$). This makes the subhalo number density comparable, for a broad range of minimal subhalo mass $m_{min}$, to the local star number density. Since our description of the subhalo population relies solely on gravitational principles, it should be valid for a wide range of cold DM candidate such as WIMPs, axions, or primordial black holes. One only needs to modify the mass function, in particular, $m_{min}$, to explore alternatives to the WIMP scenario. The resilience of subhalos increases the local WIMP annihilation rate, which, in turn, affects predictions for indirect searches. For gamma rays, we found that the boost factor was increased by at least a factor of two for $\psi < 40$ deg, and slightly less for higher values of $\psi$. The boost factor for antiprotons is also increased by a rough factor of two if subhalos are resilient to tidal disruption. For a complementary study comparing the SL17 model to simulation results regarding indirect searches in gamma rays and neutrinos, we refer the reader to Reference [99].

For future work, we plan on including a more detailed mass function, directly deriving from the primordial power spectrum, as well as the tidal heating of subhalos due to individual stars, and studying the consequences of having a large population of small objects in the Solar neighborhood.

**Author Contributions:** conceptualization, all authors; software, M.S.; investigation, M.S.; writing–original-draft preparation, M.S.; writing–review and editing, T.L. and J.L.; supervision, J.L.

**Acknowledgments:** We wish to thank M. A. Sánchez-Conde and M. Doro for inviting us to contribute to this topical review.

**Funding:** J.L. and M.S. are partly supported by the Agence Nationale pour la Recherche (ANR) Project No. ANR-18-CE31-0006, the Origines, Constituants, et EVolution de l'Univers (OCEVU) Labex (No. ANR-11-LABX-0060), the CNRS IN2P3-Theory/INSU-PNHE-PNCG project "Galactic Dark Matter," and the European Union's Horizon 2020 Research and Innovation Program under Marie Skłodowska-Curie Grant Agreements No. 690575 and No. 674896, in addition to recurrent funding by the Centre National de la Recherche Scientifique (CNRS) and the University of Montpellier. T.L. is supported by the European Union's Horizon 2020 Research and Innovation Program under the Marie Skłodowska-Curie grant agreement No. 713366. The work of TL was also supported by the Spanish Agencia Estatal de Investigación through grants PGC2018-095161-B-I00, IFT Centro de Excelencia Severo Ochoa SEV-2016-0597, and Red Consolider MultiDark FPA2017-90566-REDC.

**Conflicts of Interest:** The authors declare no conflict of interest. The funders had no role in the design of the study; in the collection, analyses, or interpretation of data; in the writing of the manuscript; or in the decision to publish the results.

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
