# Peer review of "Remnants of Galactic Subhalos and Their Impact on Indirect Dark-Matter Searches"

_galaxies, doi:10.3390/galaxies7020065_

Round 1

Reviewer 1 Report

The authors estimate the subhalo abundance in our galaxy through the use of semi-analytical techniques and they show that the boost factor for gamma rays and cosmic-ray antiprotons is increased. The manuscript is written in a good form and presents interesting results. I appreciate reading it. I only have a couple of points to ask/suggest. 1) It is not so much clear to me why they define "semi-analytical" their approach. A short note on the reasons would be useful. 2) The results are clearly galaxy-dependent. Maybe a few notes on what one expects working on other galaxies would help readers. 3) why one does not have to consider a fine-tuning issue on the choice of initial conditions performed by the authors? In which ways? 4) recently many candidates of dark matter have been excluded and proposed. A few comments on this "new era" which is now incoming would be useful. For example in: G. Bertone and T. M. P. Tait, Nature, 562, 51–56, (2018) there's a discussion on the modern problem of dark matter candidates, while in O. Luongo, M. Muccino, Phys. Rev. D 98, 103520 (2018) the authors propose a new mechanism which tightly constrains dark matter candidates. After these changes I'd like to give a final look at the manuscript

Author Response

The authors estimate the subhalo abundance in our galaxy through the use of semi-analytical techniques and they show that the boost factor for gamma rays and cosmic-ray antiprotons is increased. The manuscript is written in a good form and presents interesting results. I appreciate reading it. I only have a couple of points to ask/suggest.

Point 1: It is not so much clear to me why they define "semi-analytical" their approach. A short note on the reasons would be useful.

Response 1: we agree that this term is ambiguous in a field where people rely heavily on computer simulations. Here semi-analytical only means that we are evaluating numerically some of the integrals involved in the calculation. We added a footnote on line 31 to precise this point.

Point 2:The results are clearly galaxy-dependent. Maybe a few notes on what one expects working on other galaxies would help readers.

Response 2:our study is indeed devoted to the Milky Way only, however the subhalo model can in principle be applied to a broad class of DM objects. We have made a precision on line 56.

Point 3:why one does not have to consider a fine-tuning issue on the choice of initial conditions performed by the authors? In which ways?

Response 3:The authors are not sure what is meant by fine-tuning here. The subhalo model is flexible enough to account for different cosmological scenarios through the mass function and the mass cutoff mmin. The kinematic constraints on the Milky Way are verified by construction, so there is no free parameter to tune to agree with observations.

Point 4:recently many candidates of dark matter have been excluded and proposed. A few comments on this "new era" which is now incoming would be useful. For example in: G. Bertone and T. M. P. Tait, Nature, 562, 51–56, (2018) there's a discussion on the modern problem of dark matter candidates, while in O. Luongo, M. Muccino, Phys. Rev. D 98, 103520 (2018) the authors propose a new mechanism which tightly constrains dark matter candidates.

Response 4:we have commented on the status of DM searches and DM candidates in the introduction. The WIMP is still one of the most elegant particle candidate solving the DM issue and even though its parameter space is being probed by a number of experiments, it is not excluded yet. We have added a comment underlying the applicability of our model to other CDM candidates (axions, PBHs) on line 298, and replace “DM” by “WIMP on line 302.

p { margin-bottom: 0.25cm; direction: ltr; color: rgb(0, 0, 0); line-height: 115%; text-align: left; }p.western { font-family: "Calibri", serif; font-size: 12pt; }p.cjk { font-family: "宋体"; font-size: 12pt; }p.ctl { font-size: 12pt; }

Reviewer 2 Report

Interesting discussion about the problem of dark matter. The discussion, technically well-founded, is clear enough to give the reader a vision of the DM problem and its detection. We do not know the intimate nature of DM and any attempt to elucidate this problem is important for Cosmology.

    I only have one question, the "annihilation" of dark matter, is it at the expense of "creation" of dark energy, for instance? I would like see a brief comment on.

    So, I recommend for publication.

Author Response

Interesting discussion about the problem of dark matter. The discussion, technically well-founded, is clear enough to give the reader a vision of the DM problem and its detection. We do not know the intimate nature of DM and any attempt to elucidate this problem is important for Cosmology.

I only have one question, the "annihilation" of dark matter, is it at the expense of "creation" of dark energy, for instance? I would like see a brief comment on.

Response:annihilation of DM particles has a negligible effect on the cosmological abundance of DM today (one can check that the DM density necessary for the annihilation rate to reach the Hubble rate is far greater than the observed DM relic density). DM particles annihilate into matter and radiation. There is no direct link between DM and dark energy in the WIMP scenario.

p { margin-bottom: 0.25cm; direction: ltr; color: rgb(0, 0, 0); line-height: 115%; text-align: left; }p.western { font-family: "Calibri", serif; font-size: 12pt; }p.cjk { font-family: "宋体"; font-size: 12pt; }p.ctl { font-size: 12pt; }

Round 2

Reviewer 1 Report

accept